# In Vivo Validation of a Metacarpophalangeal Joint Orthotic Using Wearable Inertial Sensors in Horses

**DOI:** 10.3390/ani15131965

**Published:** 2025-07-04

**Authors:** Eleonora Pagliara, Federica Cantatore, Livio Penazzi, Barbara Riccio, Andrea Bertuglia

**Affiliations:** 1Department of Veterinary Science, University of Turin, 10095 Grugliasco, Italy; pagliara.eleonora@gmail.com (E.P.); livio.penazzi@unito.it (L.P.); barbararicciovet@gmail.com (B.R.); andrea.bertuglia@unito.it (A.B.); 2Pool House Equine Hospital, IVC Evidensia, Rykneld Street, Fradley, Staffordshire WS13 8RD, UK

**Keywords:** horse, metacarpophalangeal joint, orthotic, inertial sensors, rehabilitation

## Abstract

The metacarpophalangeal joint, or fetlock, plays a critical role in equine locomotion, flexing and extending with each stride. The surrounding tendons and ligament act like springs, absorbing and releasing energy and providing joint stability during motion. Injuries such as tendinitis and desmitis are common in performance horses and require complete healing to restore full functionality. While controlled exercise is fundamental to treatment, adjunctive support may be beneficial in managing severe soft tissue injuries. This study evaluated a novel fetlock orthotic device designed to limit excessive joint movement on the sagittal plane and soft tissue stretching during loading. Twelve healthy horses were assessed during walking and trotting, both without the device and with the orthotic applied at two different settings. The joint motion was quantified using sensor-based measurements. The orthotic significantly reduced the fetlock joint motion, particularly at the highest support level. These findings suggest that the orthotic device may stabilize the fetlock and serve as a supportive tool during rehabilitation following distal limb injuries in horses.

## 1. Introduction

The equine superficial digital flexor tendon (SDFT) and suspensory ligament (SL) are designed to store elastic energy, enhancing the efficiency of locomotion [1]. The cyclic load on these structures is considered the leading cause of the high incidence of disorders in equine athletes [2,3,4,5]. In racing and athletic horses, injuries to the forelimb SL, its associated structures, and the SDFT are common and potentially career-limiting [2]. Tendon lesions tend to propagate significantly after the initial injury [6]. While the size of the lesion directly impacts the prognosis, limiting the damage to prevent the spread of the original lesion appears to be crucial [7]. The biomechanical load on the SDFT is maximal during the peak extension of the metacarpophalangeal joint (MCPj) which corresponds to the midstance and its maximal palmar moment [8,9,10]. Therefore, the joint extension and the flexion as well should be ideally limited during the rehabilitation of the SDFT and SL lesions. After the initial acute inflammatory phase lasting 10–21 days [6], early mobilization stimulates intrinsic healing and collagen fiber alignment in tendon and ligament damage [11,12]. However, these structures have low tensile strength and benefit from external support for gradual loading [11,12,13,14,15].

The metacarpophalangeal joint (MCPj) is a hinge-like joint which allows movements primarily in the sagittal plane. These movements include flexion, during the swing phase of the stride and the extension, at the end of the swing, and during the stance phases of the stride [16]. The extension/flexion graph of the MCPj, which defines the kinematic curve of the movement in the sagittal plane, has a repeatable pattern across horses [17]. When walking, the stance phase extension is prolonged, showing a tendency toward two peaks separated by a slight reduction in extension. When trotting, however, there is a single cycle of extension. Both gaits present two distinct flexion peaks in the swing phase of the stride. The first flexion peak in the swing phase of the stride results from the elastic recoil of the palmar structures supporting the fetlock joint. The mechanical properties of the palmar structures of the MCPj, including the suspensory ligament (SL), sesamoidean ligaments, SDFT, and the deep digital flexor tendon (DDFT) play a crucial role withstanding the MCPj extension [18]. The extent of the MCPj extension during movement can be assessed in vivo using inertial sensors, providing a proxy measure of load distribution and strain modulation within the associated structures [19]. Wearable inertial sensor units (IMUs) are extremely versatile because they are not invasive and can be used in field conditions without the need for a standardized laboratory [20].

An orthosis is a medical device applied to the body with the purpose of providing the support, the alignment, the positioning, deformity prevention or correction, assistance to weakened musculature, or enhancement of functional capacity [21]. Orthotic devices—commonly referred to as orthoses—are increasingly employed in veterinary practice to support and protect injured limbs during rehabilitation [22].

In equine practice, orthotic devices are used to support and protect injured limbs during rehabilitation. Their primary goals are to limit joint excursion, redistribute load, and enhance compliance with controlled exercise. A MCPj orthotic (Equestride™, Marysborough House, Glanmire, Co., Cork, T45 VX26, Ireland) claims to resist the extension of the MCPj and the metatarsophalangeal joint, limiting strain on flexor tendons and the SL. This device allows for the targeted limitation of load on the damaged and surrounding soft tissues, with the load being progressively increased based on the patient’s stage of rehabilitation. Although proven effective in vitro [23], further evaluation on healthy horses is necessary to justify its use during recovery from injuries of the palmar soft tissues surrounding the MCPj. To the authors’ best knowledge, this device has never been validated in vivo in horses despite being commonly used and recommended by veterinary surgeons dealing with injured horses.

This study aims at evaluating if this MCPj orthotic effectively limits the MCPj extension (MCPj-E) and the MCPj range of motion (MCPj-ROM) in vivo. Additionally, it seeks to determine whether the device modifies the kinematic graph of the fetlock by reducing the first peak in the swing phase of the stride. We hypothesized that the device would reduce the MCPj-ROM, MCPj-E, with the highest setting being more effective than the lowest.

## 2. Materials and Methods

Horses were enrolled during routine fitness checks if they were healthy and in full training. Horses were subjectively assessed by two experienced clinicians (an ACVSMR board diplomate and a board eligible) and consensual agreement on the absence of lameness or any relevant gait alteration and any limb swelling was necessary to include the horse in the study. The horses were conveniently selected based on anatomical features to fit the device. The maximum width of the front limb metacarpophalangeal joint was between 75- and 90-mm. Ethical approval was obtained by University of Nottingham (Sutton Bonington, UK) (protocol n. 4115 240417, date of approval 2 May 2024) and the owners signed informed consent prior to the enrolment.

The device is a semirigid carbon fiber composite orthotic surrounding the metacarpal and proximal pastern region. It features two hemicircumferential cuffs connected by a system of two hinges providing structural support and controlled movement. The boot is secured to the limb using ski boot-style bindings and a foam lining that conforms to the shape of the horse’s limb. Along the palmar side of the metacarpus and the digit runs a Kevlar fiber support, anchored at both ends (Figure 1). The boot counteracts the MCPj extension using the combined strength of the carbon fiber joint and the Kevlar fibers. The resistance levels can be adjusted through three different settings, which alter the length of the Kevlar support. Setting 1 (S1) provides the least support, while setting 4 (S4) offers the maximum support. The device is available on the market for horses in rehabilitation and can be fit on the front limb and hind limb fetlocks. It claims to resist the metacarpo/metatarsophalangeal joint extension, providing support to the palmar structures of the fetlock.

Metacarpophalangeal joint angles were recorded using two inertial sensor measurements units (IMUs) (MOVIT System G1, Captiks SRL, Rome, Italy) per limb with a system previously validated against an optical motion capture system [19]. Sensors are made of a triaxial accelerometer (a full-scale range of ±2 g) and a triaxial gyroscope measuring angular velocity in the range of ±2000°/s, with a sampling rate of 200 Hz. Sensors were placed on the dorsal aspect of the metacarpus and pastern of both forelimbs (Figure 2). Continuous recording was carried out in three conditions: without the device (‘baseline’-S0) and with the device at S1 and S4 (Figure 3 and Figure 4). The standardized test was performed on a firm and even surface. The horses first got accustomed to the orthotics for 20 min with the device at setting S1 in the stable prior to obtaining the data. Before every test, the horses walked and trotted in hand for 20 min (10 min each gait) with the orthotics at the different settings. The horses then walked or trotted in a straight line starting approximately 10 m before the testing area to obtain a constant speed, which was maintained through the test. Each trial was considered valid if the horse moved at a constant pace without breaking into a different gait or looked straight ahead and if the handler did not interfere with the horse. Before each test, IMUs were calibrated following the steps provided by the manufacturers (U.S. Patent 102016000041519). Data acquisition with the IMUs was started manually before the horse entered the testing area and was stopped manually when the horse exited this area.

Accelerometric and gyroscopic raw data from each sensor were downloaded in the proprietary software (MOVIT Motion Studio 3D edition, vs. 2.3) and joint angle patterns were generated. Angle/time curves generated for each limb and condition of measurement at two gaits were analysed. The stride was manually segmented, and the MCPj range of motion (MCPj-ROM) and the MCPj extension (MCPj-E) were calculated. In detail, the MCPj- ROM was computed as the difference between the maximal and minimal fetlock joint angle recorded within the same stride. Mathematically, this was calculated as: MCPj-ROM stride = absolute value [max (fetlock joint angle stride) − min (fetlock joint angle stride)]. The MCPj-E corresponded to the minimal MCPj angle value of each segment stride (negative value). A mean of MCPj-ROM and MCPj-E was calculated based on ten strides when walking and eight strides when trotting for every limb and condition.

Data analysis was performed on R (version 4.1.2) and data were evaluated for normality and homoscedasticity. A Student *t*-test was used to compare the MCPj angle pattern of the two forelimbs of each horse at the baseline, while a one-way ANOVA was used to compare the mean value of the MCPj-ROM & MCPj-E during the three scenarios (S0, S1, and S4) for each horse. Significant results were submitted to the post-hoc test (paired *t*-test) to evaluate the comparison between the conditions. The significance was set at *p* < 0.05.

## 3. Results

This study included twelve horses: seven Irish Sport Horses, three Warmbloods, one Appaloosa, and one Welsh Cob. Their ages ranged from 6 to 20 years (mean age: 12.92 years) and their weights ranged from 515 to 678 kg (mean weight: 596.17 kg). The group consisted of nine geldings and three mares. Four horses were used for eventing: four for general riding and two for dressage; one was a hunter and one was used for endurance. All the horses were healthy and no abnormalities were detected on palpation of the forelimbs. The horses did not show any signs of discomfort due to the presence of the device and were judged to move freely after a brief initial adaptation period in which they walked and trotted in hand. No complications were encountered from the use of the devices at any time.

The average speed of the horses when walking without the supportive device was 1.49 ± 0.25 m/s while when trotting was 3.42 ± 0.83 m/s. Without the device (S0), no statistical difference was found between the left (LF) and right forelimb (RF) MCPj-ROM and MCPj-E when walking (Figure 5) and when trotting (Figure 6). In detail, the mean values of MCPj-ROM were 63.5° ± 11.4° for the LF and 62.7° ± 11.4° for the RF (*p* = 0.86) when walking. When trotting the LF MCPj-ROM was 86.2° ± 8.6° and the RF MCPj-ROM was 86.8° ± 9.8° (*p* = 0.86). The LF MCPj-E was 14.0° ± 2.1° and the RF MCPj-E was 13.3° ± 1.7° (*p* = 0.34) when walking; when trotting the MCPj-E was 26.3° ± 5.1° for the LF and 26.4° ± 45.0° for the RF (*p* = 0.96).

The MCPj-ROM was reduced when walking for S0 vs. S1 by 37% (*p* < 0.001), for S0 vs. S4 by 48% (*p* < 0.001) and for S1 vs. S4 by 11% (*p* < 0.001). When trotting, the MCP-ROM was reduced for S0 vs. S1 by 36% (*p* < 0.001), for S0 vs. S4 by 54% (*p* < 0.001) and for S1 vs. S4 by 18% (*p* < 0.001).

A significant reduction in the MCPj-E was observed for S0 vs. S1 when both walking (32%) (LF, *p* < 0.001) and trotting (32%, *p* < 0.001), for S0 vs. S4 when walking (48%, *p* < 0.001) and trotting (51%, *p* < 0.001), and for S1 vs. S4 when trotting (19%, *p* < 0.001). No significant differences were observed in the MCPj-E between S1 vs. S4 when walking for the LF (*p* = 0.208), despite being significant for the RF (*p* < 0.001). The results are summarized in Table 1.

At walk, the greatest reduction in MCPj-ROM resulted from a decrease in flexion amplitude, accompanied by a slight shortening of both the stance and swing phases (Figure 7). A slight decrease in MCPj-E and a decrease in stance phase duration were observed with both S1 and S4, with the most pronounced effect occurring in S4.

When trotting, the MCPj-E showed a substantial decrease between S0 and both S1 and S4, with the reduction being more pronounced in S4 (Figure 8). The decline in MCPj-ROM was primarily attributed to a reduction of MCPj flexion. Additionally, the first peak of flexion in the fetlock angle curve within a stride became smoother and flatter when the support device was applied (Figure 8).

## 4. Discussion

This study aims to evaluate if the MCPj orthotic would effectively limit the MCPj-E, MCPj-ROM, and modify the kinematic curve of the MCPj in vivo as the device has only been previously validated in cadaveric specimens [23]. The results of this study corroborate the hypothesis that the device significantly reduces the MCPj-ROM, MCPj-E while eliminating the first peak of flexion in the flexion/extension curve of the fetlock in horses walking and trotting in straight lines, implying a mechanical support function. This supports its use as a rehabilitation device for conditions where limiting the MCP-ROM and the MCPj-E may benefit the horses’ recovery. A reduction in fetlock hyperextension implies reduced strain on the SDFT and SL [24], which may be advantageous during the rehabilitation program following strain injuries of these structures. This orthotic device could also be used as an alternative to supporting the brace in cases of digital flexor tendons lacerations [25], joint luxation, and even fractures. The SDFT, DDFT, and SL undergo peak mechanical strain at the midstance [14,15], corresponding with the phase of the MCPj extension. During this phase, these structures function as energy-storing elements, absorbing the mechanical energy that is subsequently released through elastic recoil. During the swing phase, the MCPj exhibits two distinct flexion peaks. The first peak, occurring shortly after the toe-off, reflects an elastic response triggered by the loading of the SDFT and the SL during the preceding stance phase [9,10]. The influence of the orthotic is clearly visible in the angle-time curve of the MCPj: by providing enhanced fetlock support during the stance phase and passively restricting flexion during the subsequent swing phase, it substantially diminishes passive recoil on the SDFT. This effect is demonstrated by a notable reduction in the first flexion peak on the curve.

This present study demonstrates that the MCPj orthotic effectively attenuates the MCPj-E, thereby reducing strain on these soft tissue structures which represent an advantageous biomechanical adaptation for the horses in rehabilitation following tendon and ligament injuries. The device produced a more pronounced effect when trotting compared to walking. This observation is significant, as studies in ponies have demonstrated that digital flexor tendons strain is greater when trotting than when walking. This pattern is commonly reflected in rehabilitation protocols, where walking is built up for an extended period before introducing trotting to gradually condition the tendons and ligament and minimize the injury risk.

We hypothesized that it would reduce the MCPj-ROM and the MCPj-E, with the highest setting being more effective than the lowest. At S4, the device reduces the MCPj-ROM and the MCPj-E more effectively compared to S1 when trotting and the MCPj-ROM when walking. At S4 the device limited the MCPj-ROM and the MCPj-E when trotting and walking except for just one condition (the LF when walking) suggesting that the device could be used during different stages of the rehabilitation period and that the amount of mechanical loading on the injured soft tissue can be gradually increased under controlled conditions stimulating the healing [11,12,13,14,15]. This may lead to improved tendon healing and a quicker return to exercise in the horse.

The lack of a consistent difference in the MCPj-E between the two settings when walking suggests that S4, compared to the S1 setting, may predominantly restrict the MCPj flexion instead of the extension. The orthotic device features a Kevlar fiber support along its palmar aspect, designed to prevent the MCPj extension through the combined action of the carbon fiber joint’s compression strength and the tensile strength of the Kevlar fibers. The resistance level against the MCPj extension is adjustable via a control mechanism that modifies the length of the Kevlar tensile support [23]. Notably, the device also effectively reduces the MCPj-ROM, with shorter support lengths having a positive impact on limiting the joint range of the motion.

The device was simple to apply and remove and could be easily used by lay people. No complications or skin soreness were observed in any horse. However, injured horses in an acute or a subacute stage of tendonitis or tenosynovitis may present soft tissue swelling or synovial effusion. The application of the device could potentially cause issues that were not encountered in this study, as the horses examined were healthy. Further evaluation is necessary in lame horses but, based on our experience using the device in clinical cases, this does not appear to be a concern. Furthermore, in this study we do not consider the effects of the long-term application of an orthotic device limiting the fetlock hyperextension on the soft and hard tissues of the joint: joint stiffness, weakened ligaments, cartilage atrophy and reduced bone mass. One key consideration regarding the reduction or elimination of the elastic properties of the palmar structures surrounding the fetlock is the redistribution of internal forces within the limb.

This study has some limitations. Firstly, 3-D optical motion capture technology is considered the gold standard for kinematic analysis, of which the application requires highly standardized laboratory settings and is limited by space constraints and lighting conditions [20]. The IMU systems on the other hand are small and portable and their practicality allows their easy application in clinical settings. The IMU technology has been validated to accurately assess the sagittal fetlock joint angle, approving its use in gait analysis [19]. The device may have altered the hoof flight arc and consequently the MCPj-E and the MCPj-ROM in response to tactile stimulation on the distal limb. However, it has been previously demonstrated that horses require short periods of adaptation to tactile stimulus on the distal limb [26]. Therefore, the authors assume 20 min of walking and trotting was enough for horses to get accustomed to the device. The IMU sensors were applied on the device, not directly on to the horses’ limb. This decision was made because of the limited space on the horses’ limb after the application of the device. However, due to the orthotic’s tight fitting to the limb, independent MCPj movement was restricted, and therefore it is likely that the recorded angles directly correspond to the MCPj-ROM and the MCPj-E.

The absence of differences in the MCPj-ROM and the MCPj-E between contralateral limbs without a device (S0) confirms that the horses enrolled in this study were healthy. The device is designed to be used by horses with lame conditions. It has been demonstrated that lame horses have different sagittal fetlock joint angles compared with healthy horses, with a smaller fetlock joint range of motion both when walking and trotting [19,27]. Therefore, further studies are required to evaluate the applicability of the results of this study on lame horses with different orthopedic conditions. A different behavior of the soft tissues in the palmar aspect of the fetlock should be considered in horses having a progressive degenerative desmopathy of the suspensory ligaments, where the device should be applied to restrict the fetlock hyperextension. Further studies on this topic are necessary to justify its use in horses with this condition.

## 5. Conclusions

To conclude, this study demonstrates that this semirigid MCPj orthotic significantly reduces the MCPj-ROM and the MCPj-E during walking and trotting in healthy horses, with the highest setting (S4) providing the greatest reduction. These outcomes suggest the orthotic could be an effective tool in equine rehabilitation to mitigate excessive joint motion, support the structures of the palmar to the fetlock and aid in recovery from tendon and ligament injuries.

## Figures and Tables

**Figure 1 animals-15-01965-f001:**
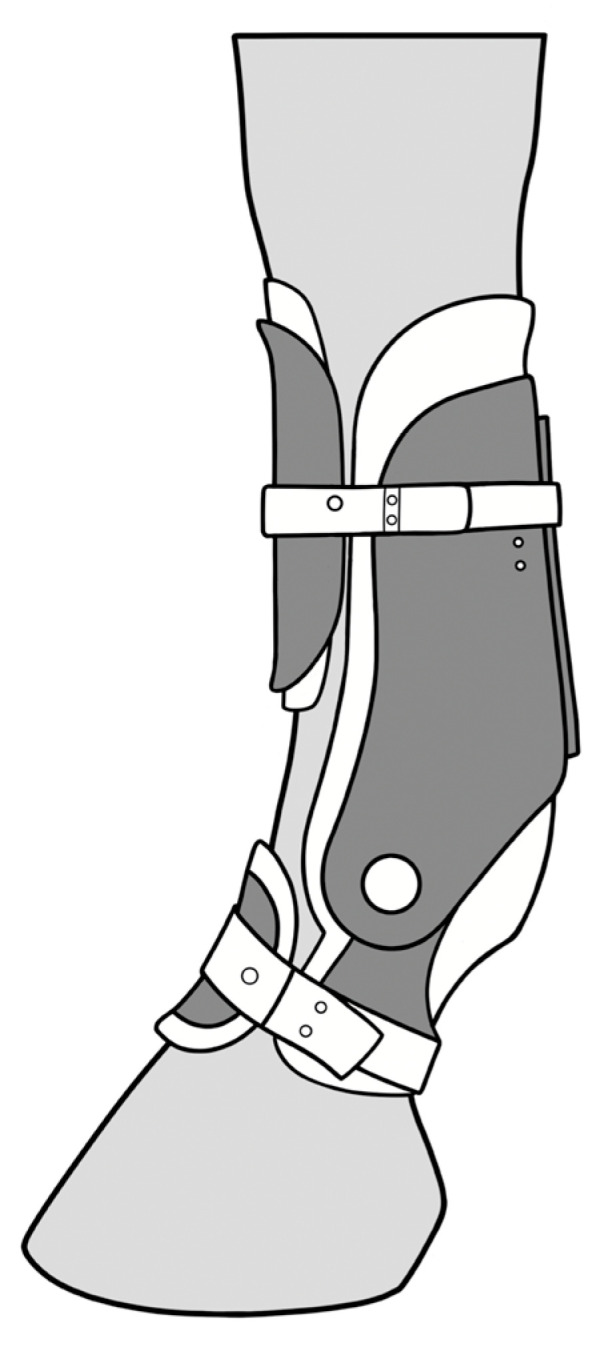
Illustrations showing the application of a metacarpophalangeal joint orthotic on a horse’s forelimb (lateral view) (image courtesy of Dr de Secondi).

**Figure 2 animals-15-01965-f002:**
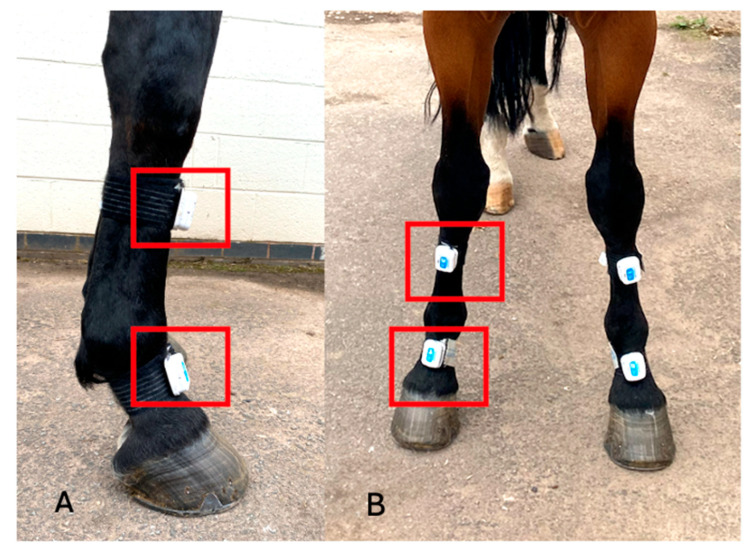
A photograph from the lateral side (**A**) and from the front (**B**) of a horse wearing the IMUs (baseline-S0) (red rectangles are used to highlight the IMUs on the right forelimb).

**Figure 3 animals-15-01965-f003:**
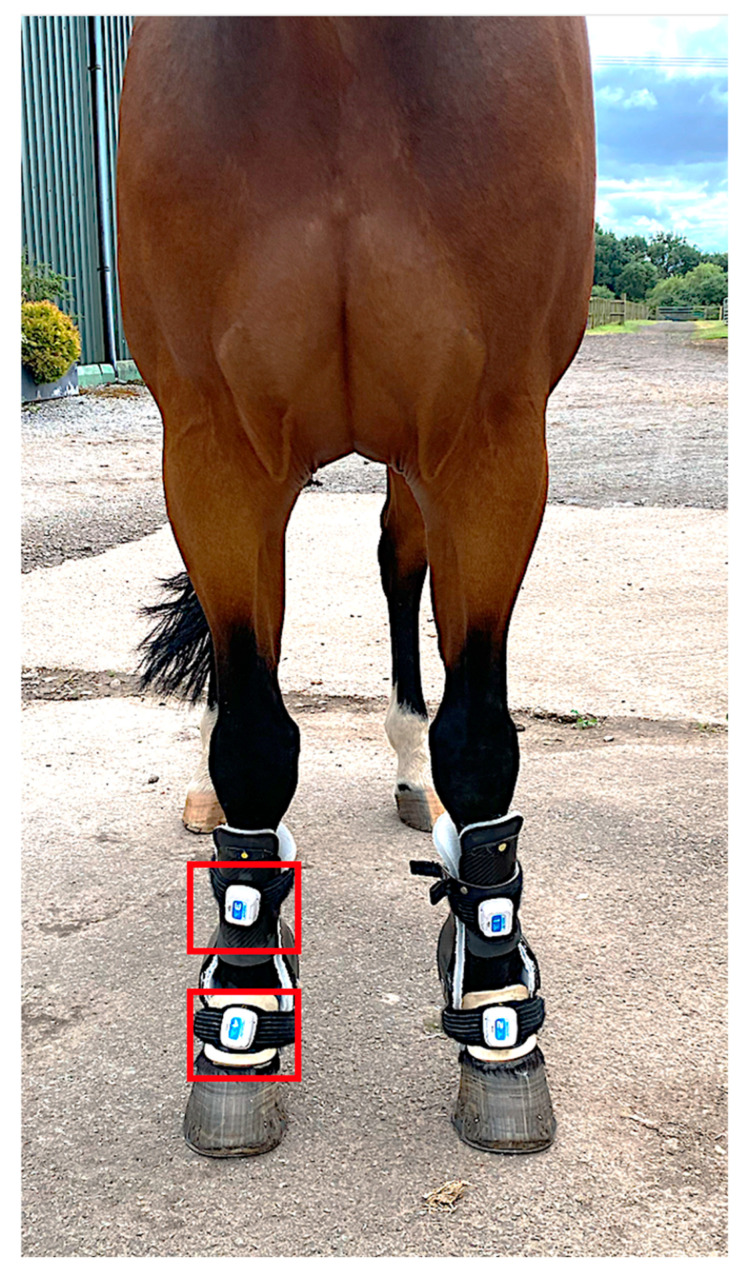
A photograph of a horse wearing the orthotic and the IMUs (red rectangles are used to highlight the IMU on the right forelimb).

**Figure 4 animals-15-01965-f004:**
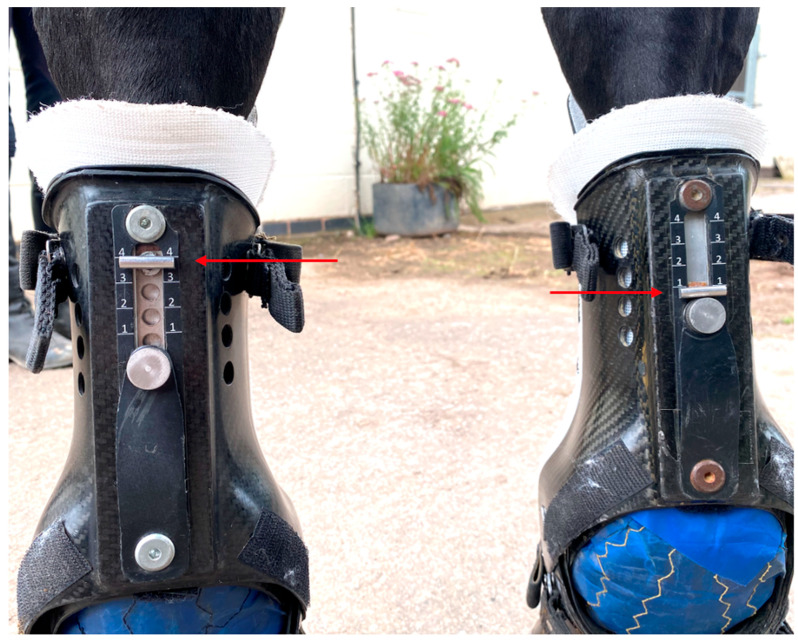
A photograph of the palmar aspect of the metacarpal region in a horse wearing the orthotic at setting 4 on the left forelimb and the device with setting 1 on the right forelimb (red arrows highlight the setting of each device).

**Figure 5 animals-15-01965-f005:**
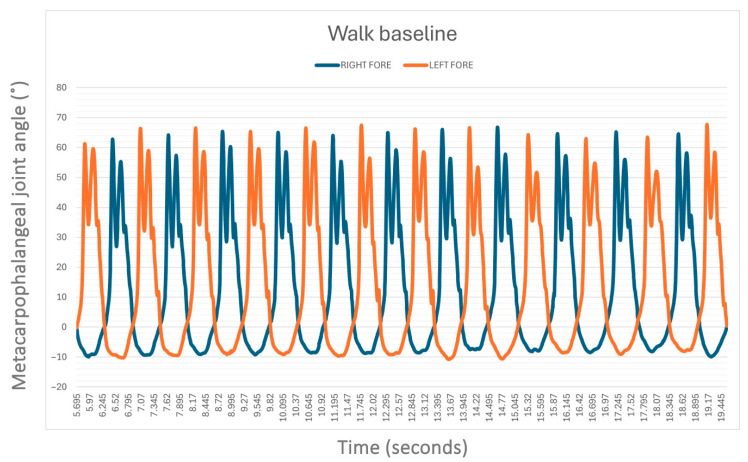
The sagittal-plane metacarpophalangeal joint (MCPj) angle-time curve in a horse (case 3) when walking without an orthotic (S0), as measured by the MOVIT IMU system. Positive angle values (°) indicate joint flexion, while negative values denote extension. The blue line indicates the right forelimb MCPj, the orange line indicates the left forelimb MCPj.

**Figure 6 animals-15-01965-f006:**
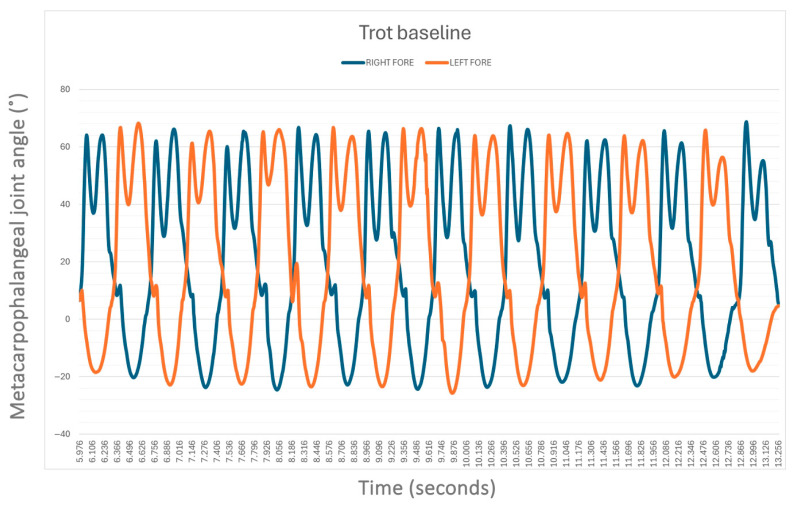
The sagittal-plane metacarpophalangeal joint (MCPj) angle-time curve in a horse (case 3) when trotting without an orthotic (S0), as measured by the MOVIT IMU system. Positive angle values (°) indicate joint flexion, while negative values denote extension. The blue line indicates the right forelimb MCPj, the orange one the left forelimb MCPj.

**Figure 7 animals-15-01965-f007:**
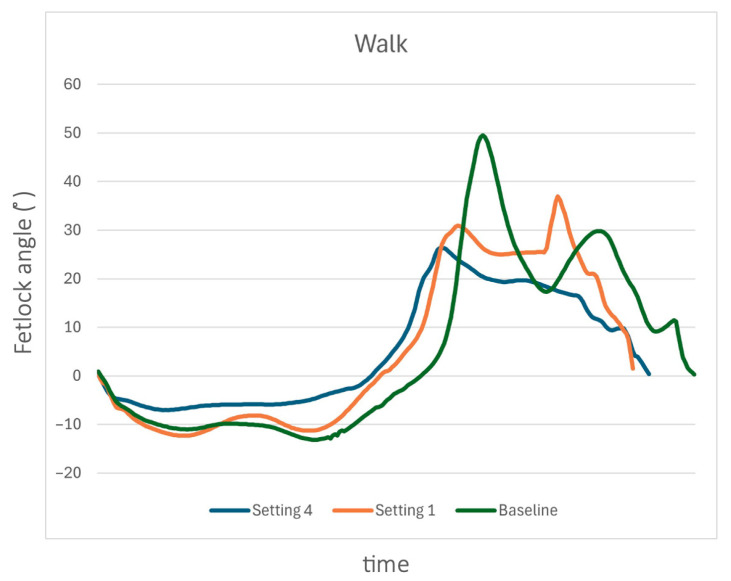
The sagittal-plane metacarpophalangeal joint (MCPj) angle-time curve over a stride when walking for a single horse (case 9) at S0 (green line), S1 (orange line), and S4 (blue line). When walking, the greatest reduction in the MCPj-ROM resulted from a decrease in the flexion amplitude, accompanied by a shortening of both the stance and the swing phases.

**Figure 8 animals-15-01965-f008:**
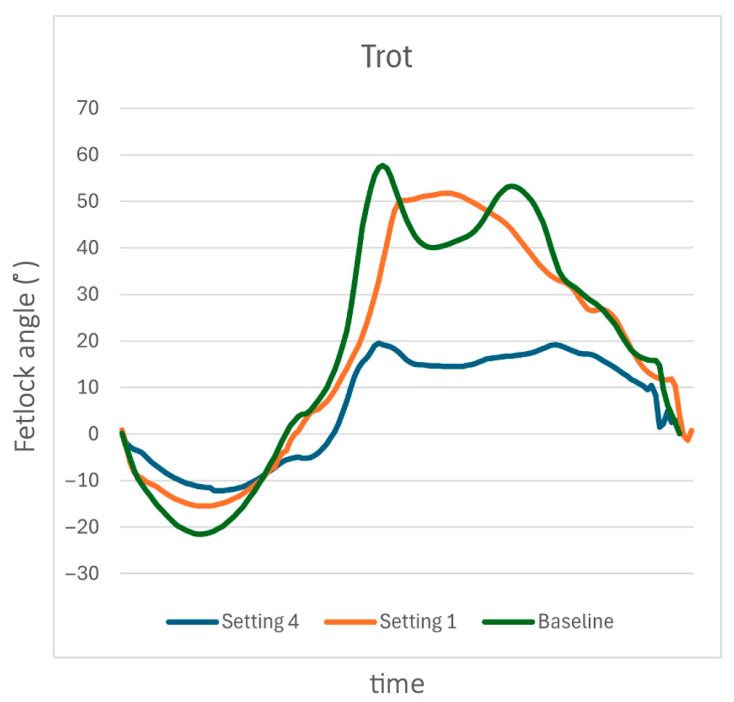
The sagittal-plane metacarpophalangeal joint (MCPj) angle-time over a stride when trotting in a single horse (case 9) at S0 (green line), S1 (orange line), and S4 (blue line). When trotting, the MCPj-E showed a substantial decrease between S0 and both S1 and S4, with the reduction being more pronounced in S4.

**Table 1 animals-15-01965-t001:** The table shows the mean values and the standard deviations of the MCPj range of motion (MCPj-ROM) and the joint extension (MCPj-E) measured without the orthotic and with the orthotic set to two different supporting settings in the forelimbs of 12 horses.

**MCPj-ROM when walking**				
**Limb**	**S0**	**S1**	**S4**	***p*-value**
LF	63.5 ± 11.4 ^a^	40.7 ± 7.8 ^b^	35.1 ± 3.5 ^c^	<0.001
RF	62.7 ± 10.1 ^a^	39.4 ± 4.6 ^b^	30.0 ± 5.9 ^c^	<0.001
**MCPj-ROM when trotting**				
**Limb**	**S0**	**S1**	**S4**	***p*-value**
LF	86.2 ± 8.6 ^a^	57.0 ± 9.0 ^b^	42.9 ± 11.0 ^c^	<0.001
RF	86.8 ± 9.8 ^a^	53.1 ± 6.5 ^b^	36.7 ± 7.4 ^c^	<0.001
**MCPj-E when walking**				
**Limb**	**S0**	**S1**	**S4**	***p*-value**
LF	−14.0 ± 2.1 ^a^	−8.6 ± 4.1 ^b^	−7.4 ± 1.8 ^b^	<0.001
RF	−13.3 ± 1.7 ^a^	−9.9 ± 3.2 ^b^	−6.9 ± 2.7 ^c^	<0.001
**MCPj-E when trotting**				
**Limb**	**S0**	**S1**	**S4**	***p*-value**
LF	−26.3 ± 5.1 ^a^	−19.5 ± 6.3 ^b^	−13.8 ± 3.9 ^c^	<0.001
RF	−26.4 ± 5.0 ^a^	−17.1 ± 3.2 ^b^	−12.2 ± 4.1 ^c^	<0.001

^a, b, c^ The mean values in the same row not sharing a common superscript letter differ (*p* < 0.05). LF: left forelimb; RF: right forelimb; S0: baseline without device, S1: device set at a lower setting; S4: device set at the highest setting.

## Data Availability

Data is contained within the article or Appendix A. The datasets generated during and/or analysed during the current study are available from the corresponding author on reasonable request.

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
