# Peer review of "In Vivo Validation of a Metacarpophalangeal Joint Orthotic Using Wearable Inertial Sensors in Horses"

_animals, 2025, doi:10.3390/ani15131965_

Round 1
Reviewer 1 Report
Comments and Suggestions for Authors
Dear Editor and Authors,
This manuscript analyzes the effectiveness of a commercially available device in reducing the movement of the fetlock joint in healthy horses. The study is scientifically well-designed and well-written. However, the objective of the study itself is not innovative, as several other studies have analyzed similar devices and reached the same conclusion: a reduction in movement.
The results of this study serve as in vivo validation of the product, which, interestingly, had not yet been done, even though it has been available for many years.
Based on the quality of the method and adequate writing, I recommend publication.
Author Response
We appreciate the reviewer’s positive feedback on the manuscript and we are grateful for the recognition of its scientific rigour. While the objective—evaluating the potential of a support device to limit fetlock joint movement—has been previously address in literature, we agree that the in vivo validation represents a novelty required to provide further evidence on the use of orthotics in horses. Hopefully the results of the manuscript will be valuable for veterinarians, trainers, and physiotherapists who rely on such devices to rehabilitate horses. Thank you for your thoughtful review and support for the publication of our work.
Reviewer 2 Report
Comments and Suggestions for Authors
1) Line 16 - consider rewording the sentence so that it is more understandable what the "3 conditions" are defined as.
2) Line 58 - consider a reference
3) Line 73 - consider a reference for the second half of the sentence
4) Figure 3 - consider improving the labels (e.g. quantify time and the degree symbol is difficult to see) and separating the two graphs on different lines or with more space.
5) Line 172 - consider labeling the results differently for MCPj-ROM to explain what to things are being compared when in brackets it has two numbers verses (vs) each other.
6) consider rewording/organising the results section from line 170 on as it is difficult to read and follow.
7) consider if Table 1 can be re-configured to make results more easily understood.
8) Line 280 - consider using terminology such as "it is likely that" or "it is unlikely that" or reference that there is no independent movement and the results strictly reflect the angles.
9) consider making it more clear from the outset and within text that the orthotic is a specific device for stabilization of the metacarpophalangeal joint. Including the title where it may be more accurate to use "metacarpophalangeal joint orthotic" instead of orthotics as you have only tested one device.
Comments on the Quality of English Language
1) Throughout the document the gramma and occasionally sentence structure needs reviewing. This includes some of the table and figure descriptions.
2) Abstract should be assessed to improve the structure of the language to improve the understanding when reading.
Author Response
Comments and Suggestions for Authors
1) Line 16 - consider rewording the sentence so that it is more understandable what the "3 conditions" are defined as.
1) The authors thank the reviewer for the comment. The sentence has been changed as suggested by the reviewer.
2) Line 58 - consider a reference
2) The authors appreciate the reviewer’s comment. The references were added as suggested by the reviewer.
3) Line 73 - consider a reference for the second half of the sentence.
3) The authors appreciate the reviewer’s comment. The references were added as suggested by the reviewer.
4) Figure 3 - consider improving the labels (e.g. quantify time and the degree symbol is difficult to see) and separating the two graphs on different lines or with more space.
4)The authors thank the reviewer for the advice. Changes have been made to the images labels.
5) Line 172 - consider labeling the results differently for MCPj-ROM to explain what to things are being compared when in brackets it has two numbers verses (vs) each other.
5) The authors thank the reviewer for the advice. Appropriate changes were made to this part of the results.
6) consider rewording/organising the results section from line 170 on as it is difficult to read and follow.
6) The authors thank the reviewer for the advice. Appropriate changes were made to this part of the results (lines 263-271).
7) consider if Table 1 can be re-configured to make results more easily understood.
7) The authors thank the reviewer for the suggestion. Table 1 has been improved.
8) Line 280 - consider using terminology such as "it is likely that" or "it is unlikely that" or reference that there is no independent movement and the results strictly reflect the angles.
8) The authors thank the reviewer for the suggestion. The sentence has been reworded as such: "However, due to the orthotic’s tight fitting to the limb, independent MCPj movement was restricted, and therefore it is likely that the recorded angles directly correspond to MCPj-ROM and MCPj-E"
9) consider making it more clear from the outset and within text that the orthotic is a specific device for stabilization of the metacarpophalangeal joint. Including the title where it may be more accurate to use "metacarpophalangeal joint orthotic" instead of orthotics as you have only tested one device.
9) The authors thank the reviewer for the suggestion. Changes have been made accordingly including the title of the manuscript. The title has been reworded as such: “In Vivo Validation of a Metacarpophalangeal Joint Orthotic Using Wearable Inertial Sensors in Horses”
Comments on the Quality of English Language
1) Throughout the document the gramma and occasionally sentence structure needs reviewing. This includes some of the table and figure descriptions.
1)Thank you for the feedback. The grammar and sentence structure have been carefully reviewed and revised throughout the document, including the descriptions of tables and figures as suggested.
2) Abstract should be assessed to improve the structure of the language to improve the understanding when reading.
2) The authors thank the reviewer for the suggestion. An English native speaker has thoroughly reviewed and revised the entire document for grammatical accuracy, clarity, and consistency in language
Reviewer 3 Report
Comments and Suggestions for Authors
The manuscript is devoted to examining a semirigid orthotic device for the metacarpophalangeal joint (MCPj) in horses. The MCPj flexes and extends during locomotion, with ligaments and tendons acting as springs to reduce energy costs. Early mobilization aids tendon( or ligament) healing, but controlled support might help in rehabilitation.
The study discovers whether the orthotic limits MCPj extension (MCPj-E) and range of motion (MCPj-ROM) while modifying joint kinematics. Inertial sensors measured MCPj movement in 12 healthy horses during walking/trotting under three conditions: no orthotic, orthotic at two settings. Authors descrbike key results: the orthotic significantly reduced MCPj-ROM and MCPj-E (p < 0.05); the higher support setting has the strongest effect; and the device demonstrates potential for rehabilitation of tendon( or ligament) injuries. The main conclusion of a manuscript is that orthotic might be an effective recovering tool for limb injuries.
The authors made a great job and manuscript is of imoptant problem. It's definetely should be published. However, I have several comments for authors.
Simple summary
rewrite SS - this is not an abstract.
The text should be shorter.
Briefly begin with joint injuries, approaches to treatment.
Then move on to the description of the method using an orthosis. Describe what you did. Write conclusive sentence
Abstract
rewrite Abstract
start with the use of orthoses in horses, for what injuries and how successfully they are used. After a couple of sentences about the metacarpophalangeal joint, types of injuries and the use of orthoses for this joint. A sentence about the purpose of the study.
Methods: Briefly what was done. How many horses were in the control and experimental groups. What injuries did the horses have. Description of statistics.
Results: what did they show. Somehow decipher the abbreviations MCPj-ROM and MCPj-E. Not all readers will understand.
Conclusions: Write a couple of sentences-conclusions.
Introduction
Rewrite the introduction.
Start with injuries and the application of orthoses in horses of different breeds and ages. Then briefly describe the principle of the joint, what problems horses encounter. How they are solved. Add the purpose of the study.
Add a list of abbreviations, it is difficult to navigate in abbreviations.
Methodology
Experimental and control groups - representatives of the same breed group and age?
Describe your groups in more detail.
Results
Add photos of orthoses on horses with light legs or adjust the brightness and contrast of the current photos. It is difficult to understand.
Make a schematic sketch of the orthosis on the limb. This will make the work more effective.
Author Response
The manuscript is devoted to examining a semirigid orthotic device for the metacarpophalangeal joint (MCPj) in horses. The MCPj flexes and extends during locomotion, with ligaments and tendons acting as springs to reduce energy costs. Early mobilization aids tendon( or ligament) healing, but controlled support might help in rehabilitation.
The study discovers whether the orthotic limits MCPj extension (MCPj-E) and range of motion (MCPj-ROM) while modifying joint kinematics. Inertial sensors measured MCPj movement in 12 healthy horses during walking/trotting under three conditions: no orthotic, orthotic at two settings. Authors descrbike key results: the orthotic significantly reduced MCPj-ROM and MCPj-E (p < 0.05); the higher support setting has the strongest effect; and the device demonstrates potential for rehabilitation of tendon( or ligament) injuries. The main conclusion of a manuscript is that orthotic might be an effective recovering tool for limb injuries.
The authors made a great job and manuscript is of imoptant problem. It's definetely should be published. However, I have several comments for authors.
A.R.: Thank you for your positive feedback and constructive comments. We appreciate your support and your recommendation for publication
Simple summary
rewrite SS - this is not an abstract.
The text should be shorter.
Briefly begin with joint injuries, approaches to treatment.
Then move on to the description of the method using an orthosis. Describe what you did. Write conclusive sentence
A.R.: The authors thank the reviewer for the suggestion. The simple summary has been revised as suggested.
Abstract
rewrite Abstract
start with the use of orthoses in horses, for what injuries and how successfully they are used. After a couple of sentences about the metacarpophalangeal joint, types of injuries and the use of orthoses for this joint. A sentence about the purpose of the study.
Methods: Briefly what was done. How many horses were in the control and experimental groups. What injuries did the horses have. Description of statistics.
Results: what did they show. Somehow decipher the abbreviations MCPj-ROM and MCPj-E. Not all readers will understand.
Conclusions: Write a couple of sentences-conclusions.
A.R.: The authors thank the reviewer for the valuable suggestion. The abstract has been revised accordingly. The abbreviations ‘MCPj-ROM’ and ‘MCPj-E’ have been removed as requested.
Introduction
Rewrite the introduction.
Start with injuries and the application of orthoses in horses of different breeds and ages. Then briefly describe the principle of the joint, what problems horses encounter. How they are solved. Add the purpose of the study.
A.R.: The authors thank the reviewer for the suggestions. The introduction has been rewritten according with the given advice. A list of abbreviations has been added in alphabetical order before the references (lines 442-454).
Methodology
Experimental and control groups - representatives of the same breed group and age?
Describe your groups in more detail.
A.R.: The authors thank the reviewer for their comment. This current study did not include experimental and control groups. As outlined in the article, horses were enrolled during routine fitness evaluations, provided they were sound and in full training. Each horse underwent a subjective assessment by two experienced clinicians. Inclusion required consensus on the absence of lameness, significant gait abnormalities, or limb swelling. Horses were conveniently selected based on anatomical compatibility with the orthotic device. Metacarpophalangeal joint angles were recorded at baseline without the orthotic (S0), and then with the orthotic at setting S1 and S4. To confirm that horses were moving symmetrically—and therefore considered sound—the MCP joint angle patterns of the left and right forelimb of each horse without device were statistically compared. Subsequently, the mean values for MCP joint range of motion (MCPj-ROM) and extension (MCPj-E) at S0, S1, and S4 were compared for each horse. Significant findings were further analysed using post-hoc testing (paired t-tests) to evaluate differences between the three scenarios (S0, S1 and S4).
Results
Add photos of orthoses on horses with light legs or adjust the brightness and contrast of the current photos. It is difficult to understand.
A.R.: The authors thank the reviewer for the suggestion. The brightness and contrast of the images was modified to improve the images’ quality. Also, some features were added to the current images to highlight IMUs.
Make a schematic sketch of the orthosis on the limb. This will make the work more effective.
A.R.: The authors thank the reviewer for the suggestion. The sketch has been added as figure 1.
Round 2
Reviewer 3 Report
Comments and Suggestions for Authors
Simple summary
The text has improved drastically< now it’s easier for nonspecialists.
The introduction to the main problem is clear. Brief method section and conclusions are clear.
Abstract
Abstract demonstrates clear improvement over version 1. Clear, readable. Still quite dense second half, and abstract might benefit from simplification. However, it might be accepted at present, the comments unnecessary to follow.
Introduction
Everything is clear. Accept, no further revision is needed. Authors did a great job!
Methods
Well- structured, everything is clear and transparent
Ethical approval is properly documented
Statistical analysis is well described, including tests and significance thresholds.
Experimental design and setup is clear.
Some sentences are quite complicated.
Accept with minor polishing.
Results
Clear description, percentage reduction highlighted
Results accompanied with p-values and multiple comparisons are made.
ANOVA followed by t-test is appropriate and well executed.
Graphical interpretation is of good quality.
Text matches the values in tables and figures.
Schematic representation is great, detailed and clear.
Some results are mentioned in text and figure legend can be trimmed.
Table 1 repeats three times, this should be deleted.
Accept
Discussion
Great quality. Robust and strong, scientifically sounds.
Accept
Conclusions
Improved Accept.
List of abbreviations
It’s great that you did it.
Overall accept at present form